# Distributed Inference and Fine-tuning of Large Language Models Over The Internet

## Abstract

Large language models (LLMs) are useful in many NLP tasks and become more capable with size, scaling to over 100 billion parameters. With the release of BLOOM-176B and OPT-175B, everyone can download pretrained models of this scale. Still, using a pre-trained 100B+ model requires high-end hardware, making it inaccessible to most researchers. Recent studies in memory-efficient training (e.g. offloading) could alleviate these costs, but they do not cover important use cases of LLMs, such as autoregressive inference. In this work, we investigate methods for cost-efficient inference of large language models, comparing local and distributed strategies. We observe that a *large enough* model (100B+) could run efficiently on geodistributed devices in a consumer-grade network, for example by connecting existing compute resources of multiple research groups or pooling under-utilized compute from multiple cloud regions. To run LLMs in this unconventional setting, we develop a fault-tolerant algorithm for inferencing language models. We propose PETALS – a decentralized system for running LLMs – and show that it can run BLOOM-176B over the Internet over $10\times$ faster than offloading for sequential generation. We evaluate the performance of our system in both simulated conditions and an actual distributed system spanning two continents. The design of PETALS allows participants to inference, and fine-tune, or inference fine-tuned models simultaneously without affecting each other's results.

## 1 Introduction

In recent years, the NLP community has found that pretrained language models greatly accelerated progress on many research problems through either fine-tuning (Radford et al., 2018) or simple prompting (Brown et al., 2020). Furthermore, their quality tends to improve as we increase model scale (Radford et al., 2019; Kaplan et al., 2020). Following this trend, modern language models often have hundreds of billions of parameters (Brown et al., 2020; Rae et al., 2021; Zeng et al., 2021; Kim et al., 2021). Most recently, several research groups open-sources their pretrained LLMs with over 100B parameters for everyone to use (Zhang et al., 2022; Khrushchev et al., 2022; Zeng et al., 2022).

Even though these models are publicly available, they are still difficult to use due to the sheer size in terms of parameters. For example, OPT-175B and BLOOM-176B need over 350GB accelerator memory for inference and significantly more for fine-tuning. As a result, even basic inference for large language models requires multiple high-end GPUs or multi-node clusters. Several recent studies propose algorithms for running large models with more affordable hardware (Pudipeddi et al., 2020; Ren et al., 2021), e.g. by offloading parameters to RAM. However, these techniques are inefficient in many practical LLM usage scenarios as we show in Section 3.1.

In this work, we search for a cost-effective way of running large pre-trained language models in their main use cases: inference, in-context learning, fine-tuning.We systematically analyze latency and throughput for training and inference tasks for these use cases and determine which factors become dominant for very large models. Notably, for models with over 100B parameters, communicating activations over a slow network can be faster than swapping layers from local RAM or SSD. Based on these observations, it should be possible to run LLMs cost-effectively by pooling together commodity hardware over the Internet.

However, existing algorithms are not designed to run inference with unreliable devices or high-latency networks. To bridge this gap, we formulate a novel algorithm for fault-tolerant distributed *inference*

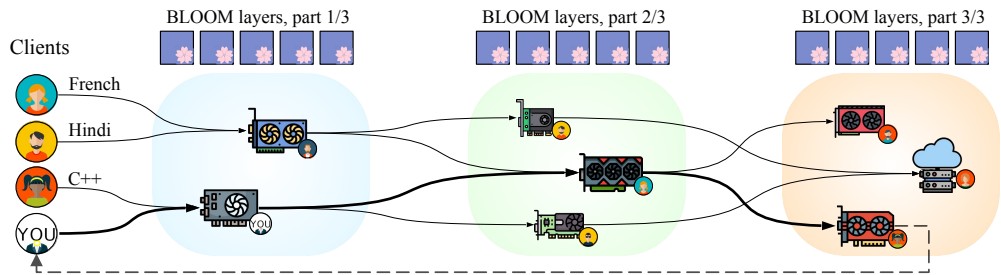

Figure 1: An overview of PETALS system. Servers store pre-trained LLM layers and temporarily hold attention caches for inferencing. Clients hold embedding layers, as well as learned prompts/adapters when inferencing fine-tuned models. Arrows denote temporary chains formed for inference.

of very large models. This algorithm takes care of inference-specific problems: keeping distributed attention caches between inference steps and recovering when some of remote devices fail or leave abruptly. This algorithm allows for several cost-effective ways of using LLMs, such as combining under-utilized GPUs in multiple cloud regions, or forming a collaboration of multiple research groups and connecting their existing infrastructure to run large models together.

The contributions of this work can be summarized as follows:

- We analyze the performance model for training very large language models and identify bottlenecks that are not addressed by existing algorithms. Notably, both local offloading and existing distributed algorithms struggle with sequential generation in different ways.

- We propose a novel distributed algorithm that can inference large (100B+) language models on distributed unreliable devices. To the best of our knowledge, this is the first algorithm that can inference LLMs with 100B+ parameters over the Internet.

- Using this algorithm, we design PETALS – a decentralized system for inferencing and fine-tuning LLMs over the Internet. When inferencing BLOOM-176B, PETALS outperforms offloading by roughly $10\times$ for autoregressive generation. The implementation of our algorithms and PETALS system is available online.[1]

## 2 BACKGROUND: EFFICIENT TRAINING AND INFERENCE

There is a wide variety of methods that can optimize training and inference for most deep learning workloads. In this section, we focus on two lines of research that are relevant for our analysis: model parallelism and parameter offloading.

### 2.1 MODEL PARALLELISM

Model parallelism is a family of distributed training algorithms that assigns each device to hold a subset of model parameters, run a subset of computations and communicate output activations. There are two main types of model parallelism: "Traditional" model parallelism and pipeline parallelism. Traditional model parallelism (or tensor parallelism) assigns each device to compute a subset of each model layer (e.g., a subset of neurons), then communicate results between each other and proceed to the next layer (Krizhevsky et al., 2012; Ben-Nun & Hoefler, 2019; Tang et al., 2020). Each device performs a symmetric computation, applied to a different slice of model weights, which makes traditional model parallelism compatible with MPI-based communication. In turn, the main performance overhead of this strategy comes from all-to-all communication (and synchronization) after each layer (Krizhevsky, 2014).

Pipeline parallelism reduces the communication overhead by assigning each device with one or several full layers (Huang et al., 2019; Narayanan et al., 2019; Yang et al., 2019). During the forward pass, each stage applies its subset of layers to the inputs supplied by the previous stage, then sends the outputs of the last layer to the next stage. For the backward pass, this process is reversed, with each pipeline stage passing the gradients to the same device that previously supplied it with input

---

[1]See github.com/iclr2023-anonymous/petals

activations. To better utilize the available devices, the pipeline must process multiple microbatches per step, allowing each stage to run in parallel on a different batch of inputs. Even with optimal execution, some of the pipeline stages will remain idle some of the time (Huang et al., 2019).

Both of these model parallel strategies are still actively used for training LLMs. However, real-world distributed training systems usually combine multiple forms of parallelism depending on hardware and network (Narayanan et al., 2021; Rajbhandari et al., 2020; Jia et al., 2019). In particular, "traditional" model parallelism is typically used within a single multi-GPU server or closely interconnected TPU cores (Narayanan et al., 2021; Shazeer et al., 2018). In turn, pipeline parallelism is used to connect multiple servers(Narayanan et al., 2021). Several recent works demonstrate that pipeline parallelism can be used for cost-efficient *pre-training* of LLMs by pooling together cheap under-utilized GPU devices (Athlur et al., 2022; Kuszmaul, 2022; Wang et al., 2022).

## 2.2 OFFLOADING

Parameter offloading relegates model parameters from accelerator memory to a slower but cheaper storage: typically RAM or SSD (Pudipeddi et al., 2020; Ren et al., 2021; Rajbhandari et al., 2021). When using the model, parameters are loaded to the accelerator just-in-time for computation, one or few layers at a time. In principle, this method allows running large models with a single low-end accelerator as long as there is enough RAM (or SSD) to store full model parameters.

The main drawback of this strategy is having to load and unload through all model parameters for each forward and backward pass, which can be time-consuming. This extra time can be amortized in workloads where model can do a lot of useful computations for each time a parameter is loaded. In practice, using offloading to run a single token through the OPT-175B on one GPU in the best-case scenario of hardware and bandwidth[2] would require 11 seconds per forward pass, or twice that for training. As we show in Section 4, real-world offloading performance is significantly worse.

Pudipeddi et al. (2020) circumvents this by training with very large batches, and hence, increasing the computation. In turn, Ren et al. (2021); Rajbhandari et al. (2021) reduce the overhead by overlapping communication and computation, that is, doing useful computation for the current layer while waiting for the transfer of the next layer to finish. However, unlike in model-parallel training, Ren et al. (2021) still requires each device to compute the full model after gathering parameters. From the efficiency perspective, the main difference with model parallelism is that, during forward pass, model-parallel algorithms need to transfer layer activations between devices, whereas offloading algorithms transfer model parameters instead.

## 3 METHOD

Using pretrained large language models for NLP tasks consists of two main workloads: inference and fine-tuning. The inference workload typically consists of encoding an input text, then generating tokens autoregressively. In turn, fine-tuning requires updating either all of the model's parameters or (more commonly for large models) a small set of trainable weights (e.g., adapters or soft prompts) by backpropagation. These two workloads also cover more advanced use cases, such as:

- Manually engineering prompts for a given task, then deploying the model with these prompts.
- Learning and inference with adapters (Hu et al., 2021; Houlsby et al., 2019; Liu et al., 2022b) or "soft" prompts (Liu et al., 2021b; Lester et al., 2021; Liu et al., 2021a).
- Distillation into a smaller task-specific model for faster inference (Schick & Schütze, 2021).

Counter-intuitively, we found that inference is more challenging than fine-tuning for cost-efficient setups. To that end, we dedicate most of this section to inference-specific problems. As for fine-tuning, we describe a generic way to support arbitrary parameter-efficient fine-tuning in Section 3.3.

## 3.1 PERFORMANCE BOTTLENECKS OF LLM INFERENCE

Unlike training, autoregressive LLM inference cannot be done with a single pass through the model. Instead, the model needs to process one token at a time, pass it through the entire model, then generate

---

[2]16-bit parameters, 350GB total, PCIe gen. 4 at 31.5GB/s (16 lanes), infinite compute and memory bandwidth.

the next token and repeat the process. In case of model parallelism, training on a sequence of $t$ tokens needs $O(n)$ communication rounds, but generating the same sequence would require $O(n \cdot t)$ rounds, making it more susceptible to network latency. Similarly with parameter offloading, generating a sequence of $t$ tokens with an $n$-layer[3] model would require loading every layer $t$ times ($O(n \cdot t)$ too).

The other problem of autoregressive generation is dealing with attention for past tokens (Vaswani et al., 2017). During an inference step $t$, each layer needs to attend to $t - 1$ past keys & values from previously generated tokens. Existing inference algorithms store past entries in accelerator memory. Caching half-precision activations of a 2048-token sequence for large models like GPT-3 (Brown et al., 2020) or OPT-175B (Zhang et al., 2022) (with 96 layers of 12288 units each) will take up 9.6GB GPU memory *for each sequence*. Offloading these cached values faces the same problems as offloading in general.

An alternative solution is to recompute all previous tokens on every inference step, storing only one set of keys & values at a time. Naturally, this approach needs quadratically more computation with sequence length $t$, for a total of $O(t^3)$ time for transformer-based models since all public LLMs with 100B+ parameters use standard attention, which has a quadratic complexity in terms of sequence length. Surprisingly, this approach is often more efficient than offloaded caching, especially for shorter sequences due to the sheer inefficiency of the latter.

Parameter offloading can still be efficient when generating *large amounts of very short sequences*. Each individual sequence still takes a long time to generate, but the system maintains high throughput by running many samples in parallel. Unfortunately, this is a niche scenario that does not cover most LLM use cases. For instance, it is incompatible with in-context learning or prompt engineering, where the model needs to process long sequences of training examples (Brown et al., 2020). More importantly, it does not support "interactive" applications where LLM needs to quickly respond to a user input. This rules out many LLM applications[4] such as conversation systems, question answering, machine translation or input completion (e.g. "Smart Compose"). To the best of our knowledge, this is a fundamental limitation of offloading that cannot be circumvented with incremental changes.

Hence, we explore a new solution based on pipeline-parallelism. Concurrent work (Aminabadi et al., 2022) uses model parallelism to inference LLMs in GPU clusters. However, their approach does not apply to our more affordable setups: cheap "preemptible" instances or connecting existing resources over the Internet. To operate in these conditions, an inference algorithm needs to deal with node preemption, network errors, and high networks latency.

## 3.2 Distributed Generation with Fault Tolerance

In this section, we formulate an algorithm for inferencing LLMs in a fleet of unreliable geographically distributed devices connected over the internet. Each device can act as a server, a client, or both. Each server holds one or several consecutive transformer blocks, while clients hold token embeddings and run inference jobs. For simplicity, we assume that every model layer is hosted on several servers and examine this assumption in the next section. Following this notation, a fault-tolerant algorithm should allow each client to complete their inference job with reproducible results even if any remote servers fail during inference.

As we discuss in Section 3.1, autoregressive generation requires many sequential communication rounds, making it sensitive to network latency. However, if every device stores its past attention cache, every communication round only transfers activations for a single token, i.e. several kilobytes of data[5]. In our algorithm, we use this model to directly minimize the inference time over all possible pipeline configurations. As we show later in Section 4.2, this allows efficient inference over a low-bandwidth Internet connection.

A more challenging problem is how to recover from node and network failures. If a remote server shuts down, any cached attention keys stored on that server will be lost with it. There are two naïve solutions to this problem: restarting inference from scratch or recomputing past embeddings on every step. Restarting might be enough at a small scale. However, running 100B+ models will involve

---

[3]In this paper, the term *model layer* (or *block*) refers to one transformer block that typically combines self-attention, a feed-forward network, normalization layers, and a residual connection (Vaswani et al., 2017).

[4]See e.g. `https://gpt3demo.com`

[5]For GPT-3 and OPT-175B, one 12288-dimensional token embedding takes up exactly 24KB.

many unreliable devices, making it unlikely to generate long sequence without at least one failure. In turn recomputing past attention caches requires communicating past tokens on every communication round, resulting in $O(n \cdot t^2)$ total data transferred, where $n$ is the number of pipeline layers and $t$ is the sequence length. In other words, both these solutions struggle to generate long sequences.

We address this problem by maintaining two types of cache: *server-side cache* holds past attention keys and values for their layers, while *client-side cache* holds past inputs sent to a given pipeline stage[6]. If a server disconnects, a client can find another server with that pipeline stage and use client-side cache to restore the server state.

The resulting procedure is described in Algorithm 1. For every pipeline stage, the client maintains a heap (priority queue) of servers that hold this stage (and may hold additional stages). The servers in queue are ordered by the network latency, measured by ping. Crucially, the queues are maintained through the lifetime of a client. To begin generation, the client runs a beam-search-like procedure to find a sequence of servers that results in the least total inference time under our performance model. When running inference steps, a client keeps track of intermediate activations sent between pipeline stages. If one of remote servers fails or leaves, the client retrieves the next best server (or multiple servers) from the heap and directs it to restore the attention state from client's cached activations.

---

**Algorithm 1** Generating sequence, client-side code

**Input:** prefix_tokens, embeddings, known_servers
```
 1: generated_sequence = list()
 2: cache = dictionary()
 3: streams = dictionary()
 4: chain = find_best_chain(known_servers)
 5: for server ∈ chain do
 6:     streams[server] = rpc_inference(server)
 7:     cache[server] = list()
 8:
 9: inputs = embeddings(prefix_tokens)
10: while should_continue(generated_sequence) do
11:     tail_servers = copy(chain)
12:     while not empty(tail_servers) do
13:         server = tail_servers.pop_left()
14:         try:
15:             ▷ Attempt normal inference
16:             outputs = streams[server].send(inputs)
17:             cache[server].append(inputs)
18:             inputs = outputs
19:         catch FailedRPC:
20:             ▷ Replace the failed server
21:             streams.pop(server).close()
22:             past_inputs = cache.pop(server)
23:             new_servers = replace_failed_server(
24:                 server, past_inputs, cache,
25:                 streams, known_servers)
26:             chain.replace(server, new_servers)
27:             tail_servers.push_left(new_servers)
28:
29:     logits = compute_logits(outputs, embeddings)
30:     next_token = choose_next(logits)  ▷ e.g. greedy
31:     generated_sequence.append(next_token)
32:     inputs = embeddings(next_token)
33:
34: for server ∈ chain do
35:     streams[server].close()
36: return generated_sequence
```

---

**Algorithm 2** rpc_inference(server)

**Input:** local_layers, stream
```
 1: cache = dictionary()
 2: for layer ∈ local_layers do
 3:     cache[layer] = make_empty()
 4: while not stream.closed() do
 5:     inputs = stream.receive()
 6:     for layer ∈ local_layers do
 7:         past_kv = cache[layer]
 8:         inputs, new_kv = forward(
 9:             layer, inputs, past_kv)
10:         cache[layer].append(new_kv)
11:     stream.send(inputs)
```

---

**Algorithm 3** replace_failed_server(...)

**Input:** server, inputs, cache, streams, known_servers
```
 1: known_servers.ban(server)
 2: missing_layers = get_layers(server)
 3: candidates = select_by_layer(
 4:     known_servers, missing_layers)
 5: chain = find_best_chain(candidates)
 6: replacements = list()
 7: while not empty(chain) do
 8:   s = chain.pop_left()
 9:   try:
10:     streams[s] = rpc_inference(s)
11:     outputs = streams[s].send(inputs)
12:     replacements.append(s)
13:     cache[s] = inputs
14:     missing_layers.pop(get_layers(s))
15:     inputs = outputs
16:   catch FailedRPC:
17:     known_servers.ban(s)
18:     candidates = select_by_layer(
19:         candidates, missing_layers)
20:     chain = find_best_chain(candidates)
21: return chain
```

---

[6]Here, a *pipeline stage* is a set of consecutive model layers hosted on one server (as in pipeline parallelism).

When a remote server fails, the algorithm needs to send $O(t)$ data (in one round) for each failed server and compute only the stages held by the failed server. This can be seen as an interpolation between naive and cached inference, depending on the server failure rate. If none of the servers fail, we recover $O(n \cdot t)$ communication, similar to Aminabadi et al. (2022). In turn, if all servers fail after one step, the algorithm effectively performs non-caching generation, which is the best option in that scenario.

In practice, it is possible to modify the algorithm to further reduce the number of network hops. In the basic formulation, all communication between pipeline stages is routed through the client, i.e. the client receives the outputs of every pipeline stage, caches it and sends it to the subsequent stage. In practice, it is more efficient to let pipeline stages communicate directly: once the server obtains output activations, it sends them to both client and the subsequent stage. This reduces the total step time since both messages are a few kilobytes in size an can be sent in parallel. To verify that both client and the next pipeline stage received the same set of activations, they can verify the checksums (i.e. hash values) of the received activations asynchronously, without blocking computation.

Algorithm 1 can support greedy inference or any sampling variants (including Holtzman et al. (2020)). However, it requires one more step to support search-based algorithms such as beam search: cache reordering. During a beam search step, a client can generate multiple continuations of the same input prefix by cloning its attention cache and dropping less likely hypotheses.

### 3.3 SYSTEM DESIGN

So far, we have focused on whether it is possible to run LLM inference on geo-distributed unreliable devices. However, this does not mean that it is practical for a researcher to do so. For example, before using Algorithm 1, each server must be assigned to a pipeline stage — and then reassigned each time another server joins or leaves the network. Here, we describe the design and implementation details that make the system practical. Our description centers around PETALS - a decentralized system that allows geographically distributed devices to run 100B+ parameter language models.

We design PETALS to run persistently across multiple projects, using a fleet of servers that can join or leave the system at any time. To operate in these conditions, a system should be able to automatically reassign layers as new servers join or leave. Furthermore, the system should support multiple clients using the model simultaneously for different tasks, e.g. when one client is inferencing the base model, another could be fine-tuning it to a different task or inferencing a previously fine-tuned task. Last but not least, it should be easy to use for popular workloads, i.e. run clients and servers.

**Server load balancing.** As we state earlier, the algorithm relies on having multiple servers per pipeline stage. To that end, each server runs a load balancing procedure to select which layers it needs to serve. Formally, servers maximize the total model throughput by choosing the blocks with the worst throughput and eliminating potential bottlenecks. Each server periodically announces its active blocks to a distributed hash table (Maymounkov & Mazieres, 2002). When a new server joins, it uses this information to identify an interval of blocks that contains most blocks with the worst throughput. This interval is always contiguous, since splitting it would harm the inference latency. Once the server has selected its layers, it measures its own throughput (both network and compute) and announces it to the distributed hash table.

Since peers may leave or fail at any time, all nodes periodically check if launching a rebalancing procedure would significantly improve the overall throughput. If it is the case, they switch layers until the throughput becomes near-optimal. In particular, if all peers serving certain blocks suddenly leave the system, this procedure quickly redistributes the remaining resources to close the emerged gaps. We provide a detailed description of the load balancing algorithms in Appendix D and validate their properties in experiments reported in Appendix E.

**Parameter efficient fine-tuning.** While LLMs achieve high quality on many problems with simple prompt engineering (Brown et al., 2020), they often need training to achieve the best results. Traditionally, this is done by fine-tuning all model parameters on the downstream task. However, for extremely large models, this strategy becomes impractical due to hardware requirements. For example, fine-tuning BLOOM-176B with Adam would require almost 3 TB of GPU memory to store the model, gradients, and optimizer states.

Fortunately, *parameter-efficient fine-tuning* methods have been developed that keep most of the pretrained model intact. Some of them choose a subset of existing parameters to update (Sung et al., 2021; Guo et al., 2021) while others augment the model with additional trainable weights (Hu et al., 2021; Houlsby et al., 2019; Liu et al., 2021b; Lester et al., 2021; Liu et al., 2021a; 2022b). Despite their lower memory requirements, parameter-efficient approaches are often competitive with full model fine-tuning (Hu et al., 2021; Liu et al., 2021a; Yong & Nikoulina, 2022) and even outperform it in low-data regimes (Liu et al., 2022a). Another appealing property of these approaches for our use-case is that they allow rapidly switching a pretrained LLM between different adapters.

By focusing on parameter-efficient fine-tuning, we are able to simplify the design of PETALS by *making clients hold all learned parameters* (see Figure 1). Servers can run backpropagation through their layers and return gradients with respect to activations, but they *do not update the server-side parameters*. Even when client communicates learned values (e.g. soft prompts) to a server, the server treats these values same as input activations. Thus, a server can simultaneously run different fine-tuning tasks without them interfering with one another. This design choice also allows PETALS users to define custom adapters in simple PyTorch without the need for network engineering expertise. We discuss the design of the client-side API in more detail for inference and fine-tuning in Appendix A.

**Memory efficiency.** Since our main intended use-case is running on inexpensive low-end devices, we need to work around their capabilities. In terms of raw FLOPs, even consumer-grade GPUs like GeForce RTX 3070 could run a complete inference step of BLOOM-176B in less than a second (NVIDIA, 2020). However, the GPU memory can only hold a small fraction of model layers: running naïvely would require 44 RTX 3070 GPUs and 44 communication rounds. To make this more efficient, we use quantization to store more parameters per GPU, reducing the number of consecutive devices and communication rounds.

More specifically, we use 8-bit mixed matrix decomposition for matrix multiplication to quantize the weights to 8-bit precision and reduce the memory footprint compared to 16-bit weights, as suggested in Dettmers et al. (2022a). This decomposition separates hidden states and weights into two portions: about 0.1% of 16-bit outlier and 99.9% of 8-bit regular values, which roughly halves the memory footprint. We verify that compressing weights does not affect model quality in Appendix B.

To send less data between subsequent pipeline stages, we use dynamic blockwise quantization (Dettmers et al., 2022b). We apply it to the hidden states before pipeline-parallel communication. Dynamic blockwise quantization halves the bandwidth requirements without any noticeable effect on generation quality. When fine-tuning, we also take advantage of gradient checkpointing (Griewank & Walther, 2000; Chen et al., 2016) and half precision to reduce VRAM usage — both of which are standard practice for large language models (Narayanan et al., 2021; Brown et al., 2020; Athlur et al., 2022). In experiments, we also apply these optimizations to baseline systems for a fair comparison.

## 4 EXPERIMENTS

### 4.1 INFERENCE WITH UNRELIABLE SERVERS

First, we conduct small-scale preliminary experiments to test the fault-tolerant generation algorithm described in Section 3.2. For these experiments, we use a smaller BLOOM model with 7.1 billion parameters (BigScience, 2022). This model consists of 30 transformer blocks with hidden size 4096. We compare our algorithm with baselines when generating a single sequence of length 512 autoregressively. For simplicity, we run all computations and communications in single precision and disregard word embeddings / logits for this set of experiments. We measure the time to run a certain number of tokens through all transformer layers. We simulate network failures by resetting pipeline stages at a certain rate.

We compare three different inference strategies:

1. Standard inference where servers store attention caches. On failure, restarts the entire generation from scratch since the failed server's caches are lost.
2. Cache-less inference that reruns past tokens on every step. On failure, restarts only the last generation step.
3. Fault-tolerant inference, as described in Algorithm 1.

Table 1: Sequential inference speed (steps/second) of BLOOM-7B1 with varying failure rates. A failure rate $p$ means that sending each set of activations to the next stage of the pipeline fails with probability $p$. The missing value means that the algorithm did not finish in a reasonable time.

| Inference Algorithm | 128 tokens, failure rate: | | | | 1024 tokens, failure rate: | | | |
|---|---|---|---|---|---|---|---|---|
| | 0 | 1e-4 | 1e-3 | 1e-2 | 0 | 1e-4 | 1e-3 | 1e-2 |
| Caching with restarts | 17.1 | 16.7 | 12 | 0.18 | 15.5 | 11.8 | 0.48 | – |
| No caching (recompute) | 3.44 | 3.44 | 3.44 | 3.44 | 0.89 | 0.89 | 0.89 | 0.89 |
| Algorithm 1 | 11.4 | 11.4 | 10.6 | 3.38 | 10.7 | 10.7 | 7.76 | 2.17 |

All runs use four pipeline stages with (8, 7, 8, 7) model layers per pipeline stage. Each pipeline stage is served by a single GeForce 1080Ti GPU; the four GPUs are running in a single system with dual Xeon Gold 6148 CPU, 12 DDR4 LRDIMM sticks with 64GB each. The system has 16 dedicated PCIe Gen.3 lanes per GPU in dual root configuration, without using PCIe switches. Each stage runs in an isolated docker containers with virtual network interfaces, but there is no limit to communication bandwidth for this experiment. We repeat all experiments 50 times and report the average time. The adjusted standard deviation never exceeds 0.2%. We use the pipeline parallelism implementation from Megatron-DeepSpeed (BigScience et al., 2022) for the stateless inference baseline.

We report all performance measurements in Table 1. Our algorithm outperforms both baselines for setups with high failure rate. Caching with restarts is most efficient for inferencing with no failures. We also note that our algorithm is somewhat slower than this baseline when there are no failures. We attribute this to a more technically advanced implementation of that baseline. Finally, non-caching inference can be competitive for short sequences (128 tokens), but slows down considerably on 1024 tokens, which agrees with out intuition from 3.1.

## 4.2 EXPERIMENTS FOR BLOOM-176B

Next, we evaluate PETALS on a more practical task of running BLOOM-176B[7] - a Transformer language model containing 70 layers with a hidden size of 14336. We evaluate three server configurations running in a network with controlled bandwidth. Our first setup consists of 3 servers, each running on an A100 80GB GPU. This is an optimistic scenario that requires the least amount of communication. In the second setup, we simulate 12 weaker devices by partitioning each A100-80GB into several virtual servers (3 large and 1 small). In total, there are 9 virtual servers running 7 blocks each, one server with 3 blocks and two more servers with 2 blocks.

We also evaluate parameter offloading, where each user runs independently on a single GPU, swapping parameters from CPU memory. We report two offloading values: the real-world throughput with DeepSpeed with the default recommended parameters, but enable pin_memory, since we found that this settings runs $1.2-2\times$ faster in all setups. For theoretical best throughput, we calculate the maximum inference and forward training throughput to receive an upper bound on offloading performance. See technical details in Appendix C. Finally, we report performance of a server with $3\times$A100 – enough GPU memory to load the entire model (the upper bound PETALS could reach). This setup runs tensor-parallel code (TP=3) based on Megatron-DeepSpeed (BigScience et al., 2022).

We evaluate PETALS with three network configurations: 1 Gbit/s with < 5 ms latency, 100 Mbit/s with < 5 ms latency and 100 Mbit/s with 100 ms latency[8]. The client-side nodes have 8 CPU cores and no GPU. We also test the effect of having multiple clients. For 12 servers with 100 Mbit/s bandwidth and 100 ms latency, if 8 clients run inference concurrently, each of them gets $\approx 20\%$ slowdown compared to the case when it runs inference alone.

In Table 2, we report the performance of sequential inference and parallel forward passes. For inference, performance does not depend much on bandwidth or sequence length but degrades in high-latency settings, especially for 12 virtual servers. In turn, training-time forward passes for large batches are affected by both bandwidth and latency. These results are shown in Table 2. We can see that offloading is about an order of magnitude slower for inference compared to PETALS. For the training-time forward pass, offloading is competitive if multiple GPUs are used and the networking for PETALS is limited to 100 Mbit/s or has high latency. In other cases, PETALS offers higher throughput than offloading for training.

---

[7]See https://huggingface.co/bigscience/bloom

[8]We set network conditions with https://github.com/magnific0/wondershaper, based on tc qdisc

Table 2: Throughput and latency for BLOOM-176B generation and forward pass

| Network | | Inference (steps/s) | | Forward (tokens/s) | |
|---|---|---|---|---|---|
| | | Sequence length | | Batch size | |
| **Bandwidth** | **Latency** | 128 | 2048 | 1 | 64 |
| Offloading on 1x A100 (best-case theoretical estimates and actual) | | | | | |
| 128 Gbit/s | – | 0.09 | 0.09 | 2.4 | 152.8 |
| 256 Gbit/s | – | 0.18 | 0.18 | 2.7 | 170.3 |
| Actual | – | 0.0485 | 0.0495 | 2.5 | 152.4 |
| PETALS on 3 physical servers, with one A100 each | | | | | |
| 1 Gbit/s | < 5 ms | 1.22 | 1.11 | 70.0 | 253.6 |
| 100 Mbit/s | < 5 ms | 1.19 | 1.08 | 56.4 | 182.0 |
| 100 Mbit/s | 100 ms | 0.89 | 0.8 | 19.7 | 112.2 |
| PETALS on 12 virtual servers, simulated on 3x A100 | | | | | |
| 1 Gbit/s | < 5 ms | 0.97 | 0.86 | 37.9 | 180.0 |
| 100 Mbit/s | < 5 ms | 0.97 | 0.86 | 25.6 | 66.6 |
| 100 Mbit/s | 100 ms | 0.44 | 0.41 | 5.8 | 44.3 |
| Same, but with 8 clients running simultaneously | | | | | |
| 1 Gbit/s | < 5 ms | 0.75 | 0.72 | – | – |
| 100 Mbit/s | < 5 ms | 0.74 | 0.7 | – | – |
| 100 Mbit/s | 100 ms | 0.36 | 0.35 | – | – |
| Entire model in GPU memory on a server with 3x A100 (upper bound) | | | | | |
| – | – | 1.35 | 1.23 | 90.6 | 286.6 |
| PETALS on 14 real servers in Europe and North America | | | | | |
| Real world | | 0.68 | 0.61 | 32.6 | 179.4 |

### 4.3 RUNNING BLOOM-176B OVER THE INTERNET

Finally, we benchmark BLOOM in a real-world distributed setting with 14 smaller servers holding 2×RTX 3060, 4×2080Ti, 2×3090, 2×A4000, and 4×A5000 GPUs. These are personal servers and servers from university labs, spread across Europe and North America and connected to the Internet at speeds of 100–1000 Mbit/s. Four of the servers operate from behind firewalls[9].

We evaluate this configuration for the same tasks as in 4.2 and report results at the bottom of Table 2. This setup is marginally slower than A100 benchmarks, which is expected due to slower hardware. However, even when communicating between different continents, PETALS maintains its efficiency, still outperforming offloading by a large margin.

## 5 CONCLUSION

In this paper, we introduced a novel fault-tolerant algorithm for inferencing large language models. On top of it, we introduced PETALS – a decentralized system for running LLMs on distributed unreliable devices connected over the Internet, which significantly outperforms other approaches to running inference on consumer-grade hardware. We demonstrated that the proposed system can scale to the largest publicly available langauge model with hundreds of billions of trainable parameters.

While our work is focused on technical aspects, running large language models over the Internet raises a broad range of related questions. One important consideration is privacy: making sure that using PETALS does not leak private data to outside peers. Last but not least, we need to ensure that participants can benefit from this system equitably, i.e. in proportion to their contribution. We discuss future problems such as privacy, security, and incentive structures in Appendix F.

---

[9]We use the Circuit Relay protocol from libp2p to traverse NATs and firewalls, see `https://docs.libp2p.io/concepts/circuit-relay/`

ETHICS STATEMENT

This work introduces a general-purpose algorithm for decentralized inference of large models, aiming to simplify access to the latest research in deep learning. Consequently, we do not envision any direct negative impacts from our research aside from granting the broader public an ability to interact with LLMs trained on uncurated web-crawled data. However, all models that can be served with PETALS are already in open access and thus can be exposed via APIs or other means: we do not release any artifacts beside the code.

One important limitation of our work in its current state is data privacy: the intermediate activations of the model for given inputs are sent to the servers without any encryption. As such, it might be possible for people hosting the servers to recover the user's input data. We discuss this limitation in more detail in Appendix F and acknowledge that the development of methods for privacy-preserving decentralized inference without performance penalties is an open research problem.

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

# APPENDIX

## A  DESIGN AND USE CASES

Practical usage of large language models can be broadly divided into two main scenarios: inference and parameter-efficient adaptation to downstream tasks. In this section, we outline the design of PETALS, showing how it handles both scenarios and also allows easily sharing trained adapters between the users of the system.

### A.1  INFERENCE OF BILLION-SCALE MODELS

When generating tokens, a client stores the model's token embeddings (which typically comprise a small fraction of the total parameter count and can fit in RAM in most modern laptops, servers, and workstations) locally and relies on servers to run Transformer blocks. Each server holds several *consecutive* blocks, the number of which depends on the server's available GPU memory. Before each inference session, the client finds a chain of servers that collectively hold all model layers.

Once the chain is formed, the client uses the local embedding layer to look up embedding vectors for prefix tokens, then sends those vectors to servers and receives new representations. Once the client obtains the outputs of the final block, it computes next token probabilities and repeats this process.

While the session is active, servers store attention keys and values from past client inputs and use them for subsequent inference steps. Clients also store past inputs to each server so that if any server fails or goes offline, another one can quickly take its place.

**Client-side API.**  To generate tokens with PETALS, one first creates an *inference session*. An inference session iteratively takes inputs as PyTorch tensors, runs them through all Transformer blocks and returns final representations as PyTorch tensors. Under the hood, sessions form server chains, hold cache, and recover from server failures in a way that is transparent to the user. An example of using an inference session is shown in Figure 2.

**System requirements.**  For BLOOM-176B inference, clients need at least 12 GB RAM, most of which is used to store 3.6B embedding parameters. We recommend at least 25 Mbit/s bidirectional bandwidth to avoid bottlenecks in network transfers. Simple greedy inference can use any CPU that runs PyTorch, but more advanced algorithms (e.g., beam search) may require a GPU.

In turn, servers need at least 16 GB CPU RAM, 100 Mbit/s bandwidth and a GPU of Turing generation or newer with at least 8 GB of memory.

### A.2  TRAINING FOR DOWNSTREAM TASKS

While LLMs achieve high quality on many problems with simple prompt engineering (Brown et al., 2020), they often need training to achieve the best results. Traditionally, this is done by fine-tuning all model parameters on the downstream task. However, for extremely large models, this strategy

```python
# Initialize distributed BLOOM model
model = AutoModelForCausalLM.from_pretrained("bigscience/distributed-bloom")
input_ids = tokenizer(prefix_text)

with model.inference_session() as session:
    # Session maintains a list of servers that remember attention KV from previous steps
    for _ in range(sequence_length):
        # Compute the word embeddings locally
        hidden = model.word_embeddings(input_ids)
        # Run distributed Transformer blocks, store attention KV for future steps
        hidden = session.step(hidden)
        # Generate the next token locally
        probs = model.lm_head(hidden)
        input_ids = sample_next_token(probs)
```

Figure 2: A basic PyTorch code snippet for sequence generation with distributed BLOOM-176B.

becomes impractical due to hardware requirements. For example, fine-tuning BLOOM-176B with Adam would require almost 3 TB of GPU memory to store model, gradients, and optimizer states.

To combat this issue, the NLP community has developed *parameter-efficient fine-tuning* methods that keep most of the pretrained model intact. Some of them choose a subset of existing parameters (Sung et al., 2021; Guo et al., 2021), others augment the model with extra trainable weights (Hu et al., 2021; Houlsby et al., 2019; Liu et al., 2021b; Lester et al., 2021; Liu et al., 2021a; 2022b).

Despite their lower memory requirements, parameter-efficient approaches are often competitive with full model fine-tuning (Hu et al., 2021; Liu et al., 2021a; Yong & Nikoulina, 2022) and even outperform it in low-data regimes (Liu et al., 2022a). Another appealing property of these approaches for our use-case is that they allow rapidly switching a pretrained LLM between different uses.

**Distributed fine-tuning.**  The core principle of fine-tuning in a distributed network is that clients "own" trained parameters while servers host original pretrained layers. Servers can run backpropagation through their layers and return gradients with respect to activations, but they *do not update the server-side parameters*. Thus, clients can simultaneously run different training tasks on the same set of servers without interfering with one another.

To illustrate this principle, we first review an example of soft prompt-tuning for text classification and then generalize it to other methods and tasks. Similarly to Section A.1, clients store the embedding layers locally and rely on servers to compute the activations of Transformer blocks. In this fine-tuning scenario, a client needs to store trainable soft prompts (task-specific input embeddings) and a linear classification head.

For each training batch, the client routes its data through a chain of remote servers to compute sentence representations, then obtains predictions with the classifier head and computes the cross-entropy loss. During backpropagation, the client runs its data through the same chain of servers in reverse order to compute gradients for the learned prompt vectors. Having obtained those gradients, the client can use a regular PyTorch optimizer to update the parameters of both the head and the prompts, then proceed to the next minibatch.

**User interface.**  To allow users greater flexibility in their training workloads, we made distributed backpropagation module compatible with the PyTorch Autograd engine. Like in the inference stage, this module handles fault tolerance and load balancing transparently to the user while allowing them to access intermediate activations and insert custom PyTorch modules. Figure 3 shows an example training code snippet.

This interface can also support other popular parameter-efficient fine-tuning algorithms, such as LoRA (Hu et al., 2021) or prefix tuning (Li & Liang, 2021). Finally, users can insert custom local modules after some of the existing blocks, which could allow use-cases like retrieval-augmented generation (Borgeaud et al., 2021; Lewis et al., 2020).

```python
# Initialize distributed BLOOM with soft prompts
model = AutoModelForPromptTuning.from_pretrained("bigscience/distributed-bloom")
# Define optimizer for prompts and linear head
optimizer = torch.optim.AdamW(model.parameters())

for input_ids, labels in data_loader:
    # Forward pass with local and remote layers
    outputs = model.forward(input_ids)
    loss = cross_entropy(outputs.logits, labels)

    # Distributed backward w.r.t. local params
    loss.backward() # Compute model.prompts.grad
    optimizer.step() # Update local params only
    optimizer.zero_grad()
```

Figure 3: A basic PyTorch code snippet of soft prompt tuning for sequence classification with PETALS.

Table 3: Zero-shot accuracy for OPT-175B and BLOOM-176B with 8-bit and 16-bit weights.

| Model | Bits | HellaSwag | LAMBADA | WinoGrande | Avg |
|-------|------|-----------|---------|------------|-----|
| OPT-175B | 16 | 78.5 | 74.7 | 72.6 | 75.3 |
|          | 8  | 78.5 | 74.6 | 71.7 | 74.9 |
| BLOOM | 16 | 73.0 | 67.2 | 70.1 | 70.1 |
|       | 8  | 72.8 | 68.1 | 70.1 | 70.3 |

Table 4: Generation throughput (tokens/s) for BLOOM-176B with 8-bit and 16-bit weights on $8\times$ A100 GPUs.

| Weights | Batch size | | |
|---------|------|------|-------|
|         | 1    | 8    | 32    |
| 16-bit  | 4.18 | 31.3 | 100.6 |
| 8-bit   | 3.95 | 29.4 | 95.8  |

### A.3 SHARING AND REUSING TRAINED MODULES

Although most fine-tuned extensions for pretrained models can be easily shared as-is, simplifying the workflow for sharing these extensions enables users to more easily adapt the model to their target scenario. Indeed, existing model hubs (Wolf et al., 2020; TensorFlow Hub; PyTorch Hub) have gained immense popularity due to many supported models and ease of use, especially when vetting different pretrained models for a given problem. One particularly relevant project is AdapterHub (Pfeiffer et al., 2020), a repository of trained adapters accompanied by a library with implementations of different adaptation methods. While PETALS does not depend on AdapterHub, it is possible to leverage this library for training adapters in the distributed setting. Instead, we support sharing modules trained by users via the Hugging Face Hub (also used as a backend by AdapterHub). Its infrastructure and the corresponding open source library simplify the learning process for users already familiar with the ecosystem. Because the primary navigation mechanism on the Hugging Face Hub are tags that have been applied to uploaded modules, a user only needs to the task it was trained on and the model upon which the adapter was built. Uploading the weights and the code of the fine-tuned module is done by committing them to a Git repository. When navigating the Hub, users can choose the most suitable adapters by filtering the list of all available modules by the required tags.

### B QUALITY AND EFFICIENCY OF BLOOM WITH 8-BIT QUANTIZATION

As shown in Table 3, this method has little effect on LLM quality for major benchmarks. In terms of inference time, Table 4 demonstrates that quantization has about $5\%$ of overhead with batch size 1 (20 tokens), but becomes negligible for larger batches.

### C ESTIMATING THEORETICAL BEST THROUGHPUT WITH RAM OFFLOADING

In this estimate, we use the best possible hardware setup for offloading: CPU RAM offloading via PCIe 4.0 with 16 PCIe lanes per GPU. In 8-bit, the model uses 1 GB of memory per billion parameters while PCIe 4.0 with 16 lanes has a throughput of 256 Gbit/s and 128 Gbit/s for PCIe 3.0. As such, offloading 176B parameters takes 5.5 seconds for PCIe 4.0 and 11 seconds for PCIe 3.0. We assume an offloading latency of zero for the upper bound estimation. In reality, many GPU servers use PCIe switches which limits the bandwidth. For instance, if two GPUs are behind a PCIe switch, transfering data to both at the same time will halve the bandwidth.

## D  DETAILS OF THE SERVER LOAD BALANCING ALGORITHMS

**Measuring throughput.**    Before joining for the first time, each server measures its Internet connection throughput (in tokens/second, using one of public web APIs for doing that) and GPU throughput (in tokens/second, using a small benchmark running several forward passes). The minimum of these values becomes the overall server throughput, which is then cached for future runs.

**Initial block assignment.**    We assume that each server holds a segment of **consecutive** transformer blocks to minimize inference latency. Clients may request to perform a forward or backward pass for the whole segment of blocks or its subsegment, if necessary. Normally, each server loads as many blocks as it can fit in its GPU memory, unless a user limits the number of blocks to utilize the rest of memory for something else.

Before starting, each server calculates the values of $t_i$ – the total throughput of servers currently holding the $i$-th block or loading it (to start holding it in a few minutes). Then, to find the best segment of blocks to serve, the server looks for the most narrow bottleneck in the network. Formally, if the model has $L$ blocks and the server can hold $K$ of them in its GPU memory, we calculate:

$$start = \underset{i=1,\, 2,\, \ldots,\, L-K+1}{\arg\min}\ \text{sorted}([t_i,\, t_{i+1},\, \ldots,\, t_{i+K-1}]) \tag{1}$$

Here, $\arg\min$ compares the sorted arrays lexicographically and chooses the leftmost $start$ in case of multiple minimums.

This way, the next joining server would always cover a block with the smallest $t_i$. If there are multiple bottlenecks like this, the server will try to cover as many of them as possible (we choose to cover the minimums first because the overall PETALS throughput is the minimum of throughputs among model blocks). Among the remaining options, we choose a segment covering as many second minimums as possible, and so on.

**Quality of block assignment.**    While we are not aware of the exact polynomial-time solution for the problem of assigning the segments optimally, we have conducted computational experiments and found out that this greedy algorithm (running in polynomial time) usually finds an assignment with total throughput of 90-100% of the optimal one (found by trying out all possible assignments in exponential time), given that the values of throughput are realistic to the PETALS setup.

**Rebalancing.**    Since servers may leave at any time, each server also periodically checks if the current assignment is "good enough" compared to the throughput estimated by running the greedy solution for servers currently present in the network.

Formally, each server periodically looks for a segment of blocks that is more appropriate than the currently loaded blocks with respect to the $\arg\min$ rule (1). If it finds one, it simulates how the rest of the servers would behave if we replace the current blocks with the new ones (how other servers would change their blocks afterwards). If the eventual throughput is at least $p\%$ better, the server commits to the change and announces that it changes the blocks, then other servers do the rest of the changes (eventually increasing the total throughput).

We use $p = 20\%$ since it gives a reasonable trade-off between the swarm throughput and the frequency of block replacements in our experiments (see Appendix E). Specifically, a lower value of $p$ leads to block replacements happening too often, which negatively affects the inference latency since each block replacement resets attention caches for this block.

**Stability of the greedy algorithm.**    The rebalancing algorithm does not cause oscillations since a series of block replacements is executed only if it leads to eventually increasing throughput by at least $p\%$. Once a "good enough" throughput is achieved, servers do not change their blocks anymore (unless an essential number of servers join or leave). We verified this behavior computationally, simulating a network with thousands of servers with different throughputs.

To conclude, this greedy heuristic allows servers to quickly close the gaps if a substantial share (up to 100%) of servers holding certain blocks leave, but avoids excess block replacements otherwise.

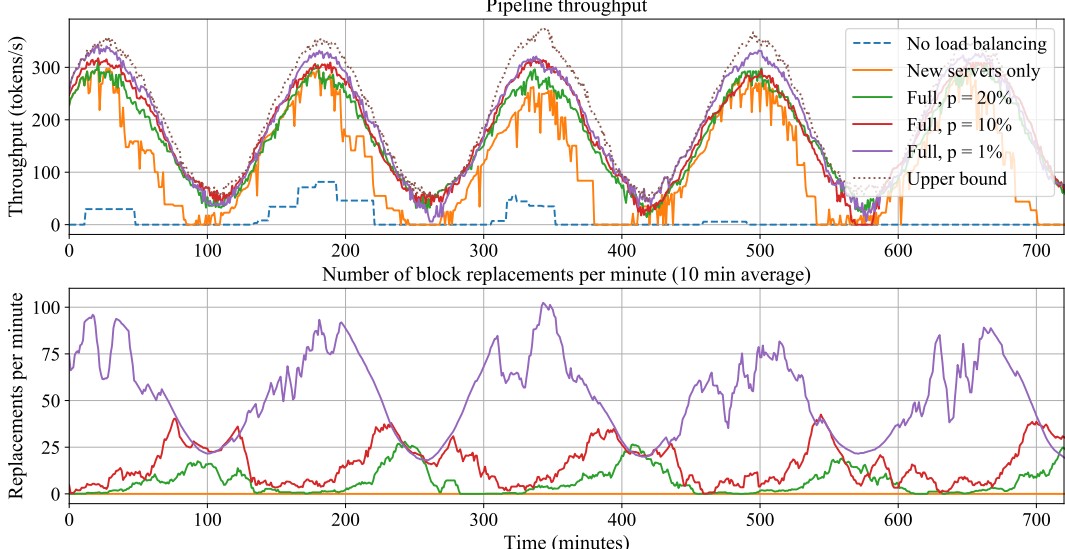

Figure 4: Behavior of the load balancing algorithms evaluated in Appendix E.

## E EVALUATION OF THE SERVER LOAD BALANCING ALGORITHMS

In this section, we measure the effectiveness of the load balancing algorithm used in PETALS. We run all experiments using a fleet of 206 virtual instances that simulate PETALS participants. To keep experiment costs manageable, we do not use GPUs for this evaluation, instead simulating uneven server throughput programmatically. For each server, we sample its throughput from the uniform distribution $t \sim \mathbb{U}[0, 100]$ tokens/second, then sample its memory size so it can hold $b \sim \mathbb{U}[1, 10]$ blocks (out of 70 blocks in total, as in BLOOM-176B).

Each server follows a certain availability schedule, i.e. turns on and shuts down at the same predefined time across all experiments. We assign these schedules such that the number of active servers follows a sine wave, simulating daily activity cycles. The schedule has approximately 100–110 active servers during peak activity and 15–25 servers at its lowest points. Note that each peak contains a different subset of 100–110 active servers out of 206 instances in total.

We evaluate the following approaches to load balancing:

1. **No load balancing** – a baseline system where servers load a random contiguous interval of model blocks.

2. **Balancing new servers only** – a simplified load balancing where servers choose the optimal blocks when joining the swarm (using the rule (1) from Appendix D) but never change them.

3. **Full load balancing** – the full algorithm, where every minute each server checks if they need to replace their blocks. We use the efficiency threshold $p$ (as described in Appendix D) to avoid excess block replacements.

4. **Upper bound** — the best-case throughput estimate that reassigns contiguous block segments to servers optimally every minute.

We report their behavior in Figure 4. The full load balancing maintains connectivity throughout the experiment and achieves throughput close to the upper bound (staying within the 10–15% range most of the time). Higher thresholds $p$ perform slightly worse during peak times but require only relatively infrequent block replacements, unlike the case with $p = 1\%$. Note that using the assignment leading to the upper bound is not possible in practice since it requires each server to load a different set of layers every minute, on top of solving the computationally expensive optimization problem.

Curiously, the baseline running load balancing for *new servers only* achieves reasonable throughput during periods where servers are actively joining. However, it quickly loses throughput when random servers leave, since this creates "bottlenecks" in the pipeline that require rebalancing of existing peers. Finally, the naive baseline with random layer assignment has zero throughput most of the time because it is unable to form a complete pipeline.

## F    DISCUSSION AND FUTURE WORK

**Privacy.**    A key limitation of our approach is that peers serving the first layers of the model can use their inputs to recover input tokens. Thus, *clients working with sensitive data should only use the servers hosted by trusted institutions that are allowed to process this data.* This limitation may be addressed in future work using secure multi-party computing (Evans et al., 2018) or privacy-preserving hardware (NVIDIA, 2022).

**Incentives for peers to contribute.**    In PETALS, peers using the client are not required to run a server. Naturally, this may lead to an imbalance between supply (peers who dedicate GPUs to serve model layers) and demand (peers using the servers to perform inference or fine-tuning for their own needs) in the network. One way to encourage users to serve model layers would be to introduce a system of *incentives*: peers running servers would earn special *points*, which can be spent on high-priority inference and fine-tuning or exchanged for other rewards. This system may be implemented using a centralized "accounting" server or in a decentralized way.

**Security.**    We assume that servers in our system are run by many independent parties. In practice, some of them may turn out to be faulty and return incorrect outputs instead of the actual results of forward and backward passes. This may happen due to a malicious intent to influence other people's outputs or, when rewards are introduced (as described above), to earn a reward for serving layers without actually performing the calculations.

A possible way to address these issues would be to use an economically motivated approach. Some servers may vouch for the correctness of their outputs (e.g., in exchange for increased inference price) by depositing a certain number of points as a pledge. Then, for each request, they announce a cryptographic hash of the input and output tensors, so anyone having the inputs can check whether the outputs are correct.

If someone finds a mismatch confirmed by a trusted third party, they can claim the server's pledge as a reward. In practice, it may be a client who suspects that they received wrong outputs or a "bounty hunter" sending requests to different servers in the hope of catching errors. While this approach still leaves a chance of receiving wrong outputs, it makes cheating costly and creates an incentive to quickly expose the malicious servers.

**Making changes to the main model.**    As discussed in Section A.2, distributed parameter-efficient fine-tuning makes it easy for users to apply the base model to new tasks. In Section A.3, we also described how these updates can be easily shared and reused by others. This capability provides a meaningful step towards *collaborative* improvement of machine learning models (Raffel, 2021): as more and more users train the base model, it will effectively become more capable over time.

Furthermore, we might expect the model parameters that perform best on a specific task to change over time. Similarly to version control systems for code, it would be useful to track versions of fine-tuned model parameters as they change. A system for rapidly testing the performance of a set of parameters on "living benchmarks" (Kiela et al., 2021; Gehrmann et al., 2022; Gao et al., 2021) would be valuable to ensure that subsequent versions improve the desired capabilities.

Apart from adaptation to new tasks, it would also be useful to eventually update the main model. Ideally, such updates could be tracked in a principled way. Users of PETALS could specify the versions of the model they want to use, and servers could indicate which versions they support. Introducing a newer version of the model then reduces to adding a new group of layers, which then naturally supersedes older parameters. Similarly, fine-tuned model adapters could be annotated with tags denoting the model version they are applicable for. Such fine-grained versioning of models is currently uncommon but would be straightforward to add to PETALS.

