# OpenReview forum: "Distributed Inference and Fine-tuning of Large Language Models Over The Internet"
_ICLR.cc/2023/Conference — Submitted to ICLR 2023_

### Official Review · Reviewer_s5Ah · 2022-10-25

**Confidence:** 3
**Correctness:** 3
**Technical Novelty And Significance:** 3
**Empirical Novelty And Significance:** 3
**Recommendation:** 5

**Clarity, Quality, Novelty And Reproducibility:**

The clarity, quality, and reproducibility are good.
Regarding novelty, it might borrow part of the contents from the existing research on fault tolerance in distributed systems.

**Details Of Ethics Concerns:**

There is a data privacy issue when sending intermediate activation to other works. It has been discussed in details in the paper.

**Strength And Weaknesses:**

Strength:
1. Firstly, the paper is generally well-written and easy to follow.
2. The observation that transferring activation is faster than offloading weights in LLMs is interesting.
3. The proposed method addressed an important question and allowed communities with lower-end GPUs to be able to serve and tune LLMs. There could potentially be a new business model based on collective inference.

Weakness:
1. I am not very familiar with distributed system research. But from my understanding, fault tolerance in a distributed system has already been well studied. It seems the proposed method only applies the techniques to LLM inference and does not have a significant novelty.
2. From my understanding, the method can only work when the model is large enough (as the authors mentioned, >100B). Could you please show how it works for smaller LMs? What model size is the turning point?
3. When comparing the baseline of offloading, did you apply the same quantization technique to reduce bandwidth requirement?
4. Did you implement pipeline parallelism to further improve the throughput?


**Summary Of The Paper:**

In this paper, the authors proposed a method to perform inference na fine-tuning of LLMs over in a distributed manner over the internet, based on the fact that activation communication between geo-distributed devices could be faster than offloading to local memory for really large LMs. The authors developed a fault-talent algorithm to handle the error and proposed some system design guidelines. Experimental results show that the proposed method can outperform caching/recomputing under higher failure rates and has a higher throughput than RAM offloading.

**Summary Of The Review:**

Please see the Strength and Weakness section. I am not 100% familiar with the literature on distributed systems. And I am open to raising scores given a convincing reply.

---

> ### Author Response · Authors · 2022-11-10
> **Response to Reviewer s5ah**
>
> We thank the reviewer for their feedback and address their comments below.
>
> > 1. [...] from my understanding, fault tolerance in a distributed system has already been well studied. It seems the proposed method only applies the techniques to LLM inference and does not have a significant novelty.
>
> We respectfully disagree. As we review in Section 3.1, LLM inference presents a special case where existing fault-tolerant training algorithms are inefficient. Since LLM inference is an important problem, we expect the first efficient fault-tolerant algorithm for LLM inference to be a valuable and novel contribution.
>
> > 2. From my understanding, the method can only work when the model is large enough (as the authors mentioned, >100B). Could you please show how it works for smaller LMs? What model size is the turning point?
>
> The turning point depends on many factors: the GPU type, network speed, and cpu-gpu interconnect. Empirically, on a machine with RTX 3080 GPU, our approach is the fastest for models with 13B+ parameters, offloading is the fastest for 10B-13B models, and models under 10B can fit in the GPU memory. In case of A100 GPU with 80GB memory, PETALS turns out to be the fastest option for 90B+ models, while offloading is the fastest for the 80B-90B range. In either case, PETALS is the fastest option for most existing models that do not fit in a single GPU.
>
> > 3. When comparing the baseline of offloading, did you apply the same quantization technique to reduce bandwidth requirement?
>
> We always use the same set of optimizations for Petals and the baseline algorithms, as mentioned in the last paragraph of Section 3.3. In particular:
>
> - In Section 4.1 (small model), neither algorithm uses quantization (see the first paragraph).
> - In Section 4.2, both PETALS, DS-Offload and our best case estimates use 8-bit quantization.
>
> > 4. Did you implement pipeline parallelism to further improve the throughput?
>
> When training or inferencing with batch size greater than 1, PETALS behaves similarly to pipeline parallelism when different sequences are processed in parallel. Specifically, **(1)** when a batch of sequences progresses to the second stage, the first stage can start processing another batch simultaneously, **(2)** two sequences can be processed by two parallel servers if they are available, and **(3)** a single server can process workloads from different clients. We will be happy to provide further details if requested.

---

### Official Review · Reviewer_X2vi · 2022-10-27

**Confidence:** 4
**Correctness:** 2
**Technical Novelty And Significance:** 3
**Empirical Novelty And Significance:** 3
**Recommendation:** 5

**Clarity, Quality, Novelty And Reproducibility:**

The paper is generally well written and easy to read. The paper has several novel aspects and interesting technical details. The evaluation could be improved as described earlier.

**Strength And Weaknesses:**

Although, pre-trained LLMs are now freely available for download the hardware cost of inference and finetuning computation is still quite prohibitive and is a barrier to broad accessibility of AI advancements. Thus, the paper is tackling an important problem. The proposal of a community-approach that involves pooling consumer-grade resources together to inference and finetune LLMs is also reasonable and applicable to a broad class of users.  The presented algorithm and implementation details are useful in understanding some of the technical challenges.

My main concern is the evaluation, which I feel does not present a complete and fair assessment of PETALS. Below are specific examples.

1. Overhead of fault tolerance (4.1)
    a) What are the runtime and memory costs of client and serving caching, and how they scale with increasing client and server counts?
    b) When clients don't have GPUs, does this mean client-server communication involve GPU/CPU copying on the server?
    c) Are the runtime and memory costs of fault tolerance reflected in the results in 4.2 and 4.3?
    d) Table 1 only shows fault tolerance evaluation for inference. Is the fault tolerance provided for fine tuning?
    e) In Table 1, PETALS is the best option in only 2 of the 8 scenarios. Caching with restarts wins 5 times, No caching wins once. This is not very convincing for PETALs.

2. Comparison to Offloading (4.2)
   a) Using 1 A100 for Offloading and 3 A100 for PETALs does not seem like a fair comparison. Can you elaborate?
   b) The theoretical latency numbers of Offloading suggest that 16-bit weights are being used rather than 8-bit weights, is this correct?
   c) Are the A100 GPUs connected by PCIe or NVLINK?
  d) Depending on the response to the preceding 3 issues, a fair evaluation could mean scaling the Offloading performance by up to 18X:
    3X for GPU count, 3X for parallelizing weight fetching from CPU, and 2X for 8-bit weight quantization.
  e) Why are there no forward pass results for the multiple client section of Table 2?


**Summary Of The Paper:**

The paper proposes PETALS, a system that enables inference or finetune processing of large language models (LLM) without the cost of HPC hardware, but rather by sharing a large pool of consumer-grade hardware. PETALS uses pipeline parallelism to partition LLM layers over the geographically distributed devices and proposes a fault tolerant algorithm for inference and fine tuning to address reliability issues. The paper presents some evaluation of the fault tolerance algorithm and performance comparison to offloading.

**Summary Of The Review:**

While I think PETALS would be a useful system to many users, the current evaluation makes it difficult to understand how well it would function and perform in real life scenarios. Moreover, the previously noted evaluation issues makes it difficult to know the scenarios that PETALS would be preferable to caching with restarts and offloading.

---

> ### Author Response · Authors · 2022-11-10
> **Response to Reviewer X2vi (Part 2 of 2)**
>
> > Q2. a) Using 1 A100 for Offloading and 3 A100 for PETALs does not seem like a fair comparison. Can you elaborate?
>
> It would indeed be unfair to compare the same workload on 1 vs 3 GPUs. However, this is not what we compare in the paper. Our main hypothesis is whether multiple users can run BLOOM faster by **collaborating together** - as opposed to running the model independently.
>
> In other words, if there are 3 users with local GPU servers, we compare two settings:
>
> 1. Each user runs their task independently on their own GPU, without communicating with others (requires offloading).
> 2. Users communicate over the Internet and collectively run all tasks using Petals.
>
> We argue that this is a reasonable way to test our hypothesis, since both strategies do the same computation per GPU. In Table 2, we show that PETALS outperforms the first strategy even when 3 servers run 8 tasks (i.e. there are 5 “freeloaders” in the system).
>
> > Q2. b) The theoretical latency numbers of Offloading suggest that 16-bit weights are being used rather than 8-bit weights, is this correct?
>
> That is not the case. To sanity-check this, consider the time to load the entire model in 16-bit precision over a 128 Gbit/s bus:
>
> ${{176 \cdot 10^9 \texttt{ parameters } \times \texttt{ } 16 \texttt{ bits }} \over {128 \cdot 10^9 \texttt{ bits per second }}} = 22 \texttt{ seconds }$
>
> In contrast, our theoretical estimate for that setting is $\approx 11.1$ seconds per step (from 0.09 tokens per step), which is roughly twice as fast, as expected for 8-bit weights.
>
> > Q2. c) Are the A100 GPUs connected by PCIe or NVLINK?
>
> In all our experiments, the GPUs communicate using TCP (i.e. network), since they are located on different servers. Inside each server, the GPU is connected to the rest of the system by PCIe.
>
> > Q2. d) Depending on the response to the preceding 3 issues, a fair evaluation could mean scaling the Offloading performance by up to 18X
>
> Based on our previous responses, we argue that our estimate remains valid, but we are open to further discussion.
>
> > Q2. e) Why are there no forward pass results for the multiple client section of Table 2?
>
> Since the forward pass deals with multiple parallel sequences in a batch, our implementation already runs computations in parallel. Hence, we expect near-linear scaling.
> If requested, we can conduct these experiments and report them in the nearest revision.
>
> __Edit:__ We swapped Parts 1 and 2 of the response, so that Part 1 is displayed first on OpenReview.

---

> ### Author Response · Authors · 2022-11-10
> **Response to Reviewer X2vi (Part 1 of 2)**
>
> We thank the reviewer for their interest and answer their questions below. We will be happy to answer further questions if requested.
>
> > Q1. a) What are the runtime and memory costs of client and serving caching, and how they scale with increasing client and server counts?
>
> In terms of runtime, both caching schemes keep specific intermediate activations after they are computed. They do not require additional GPU computation / communication and incur a O(1) overhead for updating the metadata. In practice, this does not affect the wall time since the update happens in parallel with asynchronous gpu computations or networking.
>
> The memory requirements for client-side caching is $2 \cdot t \cdot h \cdot s$ bytes, where $t$ is the sequence length (tokens), $h$ is the hidden size (units) and $s$ is the number of active pipeline stages used by this client, no more than one per layer. The factor 2 is due to 16-bit precision. In turn, server-side caching requires $4 \cdot t \cdot h \cdot b$ bytes of memory (server-side), where $b$ is the number of blocks held by this specific pipeline stage. The factor 4 is from storing both keys and values in 16-bit precision. Note that this cache size is standard for transformer-based models.
>
> > Q1. b) When clients don't have GPUs, does this mean client-server communication involve GPU/CPU copying on the server?
>
> In our implementation, the server always needs to copy activations from host to device, regardless of the client type. However, the GPU/CPU bandwidth (PCIe) is over 100 Gbit/s, while the network bandwidth is at most 1 Gbit/s in all our experiments. In practice, we found that this GPU/CPU transfer does not have a significant effect on the total inference speed. However, in principle, one could avoid the host-to-device copy using GPUDirect RDMA (if your hardware supports it).
>
> > Q1. c) Are the runtime and memory costs of fault tolerance reflected in the results in 4.2 and 4.3?
>
> They are reflected in Section 4.3 (real world scenario), but not in Section 4.2, as these experiments were run in a local network.
>
> > Q1. d) Is the fault tolerance provided for fine tuning?
>
> Yes. Since fine-tuning does not maintain any attention caches between forward/backward passes, PETALS simply reruns failed requests using a different server (see supplementary code src/client/sequential_autograd.py). In the paper, we omit this for brevity to focus on the more challenging generation task. We will be happy to discuss this further if the reviewer has further questions.
>
> > Q1. e) In Table 1, PETALS is the best option in only 2 of the 8 scenarios. Caching with restarts wins 5 times, No caching wins once. This is not very convincing for PETALs.
>
> Our experiments aim to explore the limitations of the proposed approach, not just to advertise strengths, so we deliberately chose an unfavorable setup. As we describe in Section 4.1, the baselines run the small model locally on GPU, while PETALS performs actual network communication (TCP). A more practical evaluation setup can be found in Sections 4.2 and 4.3, where we consider large models PETALS was designed for.

---

### Official Review · Reviewer_JnBu · 2022-10-30

**Confidence:** 4
**Correctness:** 2
**Technical Novelty And Significance:** 2
**Empirical Novelty And Significance:** 3
**Recommendation:** 5

**Clarity, Quality, Novelty And Reproducibility:**

The paper is well-written and easy to understand in general. With the provided code, I believe the results of the paper can be reproduced by researchers or engineers with access to geo-distributed devices. However, I have concerns on the novelty of the paper in the weakness discussion above. Below are some of my extra questions to the work:

1. Page 3: “Counter-intuitively, we found that inference is more challenging than fine-tuning for cost-efficient setups.” Is this due to inference is an autoregressive iterative decoding process? Please elaborate.
2. Page 7: “Servers can run backpropagation through their layers and return gradients with respect to activations”: During fine-tuning, how does the server deal with the intermediate activation from the forward pass? Do you always turn on gradient checkpointing, or do you cache the intermediate activation during the forward pass?
3. Table 1: How is failure rate defined exactly?
4. Table 1: Why does Algorithm 1 perform worse than recompute with 128 tokens and 1e-2 failure rate?
5. Do you only focusing on the case where there is only one client? How do you handle multiple clients?

Nit:

Page 1: “they are still difficult use due to” → “they are still difficult *to* use due to”

**Strength And Weaknesses:**

## Strength

1. The paper proposes a valid combination of multiple techniques for efficient inference and fine-tuning of LLMs, including pipeline parallelism, adapters, quantization and compression.
2. The engineering behind the system is non-trivial and I appreciate the authors for open-sourcing their work. I believe the work will be beneficial for researchers with access to geo-distributed machines.

## Weakness

1. Pipeline parallelism is a well-studied technique for large language models. The way this paper applies pipeline parallelism is not new compare to previous works.
2. The dynamic load balancing (i.e., schedule which layers to run on each GPU) are not evaluated in the experiments. How does this algorithm adapts to different hardware configurations?
3. Comparing the proposed setting with the offloading techniques is not a completely fair comparison: The offloading setting only has a single GPU, but the geo-distributed setting has many more GPUs and thus much higher total computational power.
4. A more important comparison is to compare the efficiency of this method with the setting where the GPUs are within a single datacenter and the optimal performance in that case.

**Summary Of The Paper:**

This paper presents Petals, a system for inference and fine-tuning of large language models on distributed machines over the internet. The system mainly includes the following techniques:

- Pipeline parallelism across geo-distributed devices.
- Fault tolerance via storing an extra copy of pipeline communication data within the client.
- A dynamic load-balancing algorithm that eliminates performance bottlenecks.
- Using adapters for fine-tuning and storing the adapter parameters in the client for fault-tolerant fine-tuning.
- Quantization and compression to reduce memory footage.

The authors evaluate the system on a smaller BLOOM model for fault tolerance performance and test on the large BLOOM-175B model with multiple hardware settings to test the latency of the proposed system.

**Summary Of The Review:**

The paper proposed a solid system for geo-distributed inference. The work includes solid system engineering but lacks enough novelty for a research paper.

---

> ### Author Response · Authors · 2022-11-10
> **Response to Reviewer JnBu (Part 2 of 2)**
>
> ### Answering questions
>
> > Q1: [on “inference is more challenging than fine-tuning”] Is this due to inference is an autoregressive iterative decoding process? Please elaborate.
>
> While training, transformers can process all tokens in parallel, accessing each model parameter once per training sequence. In turn, LLM inference is inherently sequential: one needs to finish generating the $i$-th token before generating the $(i+1)$-th one. This requires loading each parameter once for every token in the generated sequence. Thus, compared to LLM training, inference is more sensitive to memory bandwidth and network latency, and less sensitive to the raw compute FLOP/s.
>
> As a result, many distributed *training* algorithms become inefficient when reused for inference workloads. For a more detailed explanation, please see the beginning of Section 3.1 (end of page 3).
>
> > Q2: Do you always turn on gradient checkpointing, or do you cache the intermediate activation during the forward pass?
>
> We always use gradient checkpointing, which is a standard choice for pipeline-parallel LLM training. Both GPipe [2] and Megatron [3] rely on gradient checkpointing to fit more micro-batches in VRAM.
>
> [2] Huang, Yanping, et al. "Gpipe: Efficient training of giant neural networks using pipeline parallelism." Advances in neural information processing systems 32 (2019).
>
> [3] Narayanan, Deepak, et al. "Efficient large-scale language model training on gpu clusters using megatron-lm." Proceedings of the International Conference for High Performance Computing, Networking, Storage and Analysis. 2021.
>
> > Q3. Table 1: How is failure rate defined exactly?
>
> Every time a user sends activations to a remote server, there is a chance that the remote server will fail instead. When generating 128 tokens with 4 consecutive pipeline stages, there are $4 \times 128 = 512$ spots where a failure may occur. When the algorithm switches to a new server (pipeline stage), it does not automatically have all past attention keys/values of the failed stage. We will clarify the failure rate definition in the nearest revision of the paper (by the end of Monday AOE).
>
> > Q4. Table 1: Why does Algorithm 1 perform worse than recompute with 128 tokens and 1e-2 failure rate?
>
> We attribute this to implementation details. We designed PETALS for running very large (100B+) language models. For a relatively small 7.1B model, the overhead for TCP connections is still quite significant. In contrast, the recomputation baseline is a PyTorch script that does not use networking for simplicity. Please let us know if you'd like us to perform a more detailed investigation.
>
> > Q5. Do you only focusing on the case where there is only one client? How do you handle multiple clients?
>
> We focus on the use case where there are multiple clients (e.g. Table 2 has a section with 8 clients per 3 users). This is crucial, since we intend PETALS to provide value to all active participants. From a technical standpoint, each independent client opens a streaming RPC to each pipeline stage. In turn, a server (pipeline stage) maintains a separate attention cache for every client.
>
> When generating a sequence, clients naturally alternate between using different pipeline stages, e.g. client A uses stage 2 of 4, while client B uses stage 1 of 4 in parallel. We use the first-come-first-served policy when multiple clients access the same pipeline stage simultaneously. We will happily provide more implementation details if requested.
>
> We also thank the reviewer for finding the typo and will fix it promptly.
>
> __Edit:__ We swapped Parts 1 and 2 of the response, so that Part 1 is displayed first on OpenReview.

---

> ### Author Response · Authors · 2022-11-10
> **Response to Reviewer JnBu (Part 1 of 2)**
>
> We thank the reviewer for their feedback and address their comments below.
>
> ### Addressing concerns
>
> > 1. The way this paper applies pipeline parallelism is not new compared to previous works.
>
> While our algorithm shares some communication patterns with pipeline parallelism, it is an entirely different parallelization scheme that goes much beyond pipeline parallelism. This is the first work that allows distributed inference of LLMs outside of HPC clusters, e.g. using unreliable machines connected over the Internet. Our work requires novel distributed inference algorithms specifically to operate in these conditions. The only other (concurrent)  work [1] that studies distributed generation needs reliable high-interconnect GPU clusters.
>
> [1] Aminabadi, Reza Yazdani, et al. "DeepSpeed Inference: Enabling Efficient Inference of Transformer Models at Unprecedented Scale." arXiv preprint arXiv:2207.00032 (2022).
>
> > 2. How does (load balancing) adapt to different hardware configurations?
>
> We will include load-balancing experiments to the updated version of the paper (expected by the end of Monday AOE).
>
> > 3. Comparing the proposed setting with the offloading techniques is not a completely fair comparison (1 vs 3 GPUs)
>
> It would indeed be unfair to compare the same workload on 1 vs 3 GPUs. However, this is not what we compare in the paper. Our main hypothesis is whether multiple users can run BLOOM faster **by collaborating together** - as opposed to running the model independently.
>
> In other words, if there are 3 users with local GPU servers, we compare two settings:
>
> 1. Each user runs their task independently on their own GPU, without communicating with others (requires offloading).
> 2. Users communicate over the Internet and collectively run all tasks using Petals.
>
> We argue that this is a reasonable way to test our hypothesis, since both strategies do the same computation per GPU. In Table 2, we show that PETALS outperforms the first strategy even when 3 servers run 8 tasks (i.e. there are 5 “freeloaders” in the system).
>
> > 4. A more important comparison is to compare the efficiency of this method with the setting where the GPUs are within a single datacenter and the optimal performance in that case.
>
> We agree and will provide this comparison in the nearest update (expected by the end of Monday, AOE).

---

### Official Review · Reviewer_3oHu · 2022-10-30

**Confidence:** 4
**Correctness:** 2
**Technical Novelty And Significance:** 3
**Empirical Novelty And Significance:** 3
**Recommendation:** 6

**Clarity, Quality, Novelty And Reproducibility:**

I really enjoyed this paper.  It's a great problem and creating a system for inference/fine-tuning on LLMs would be useful.   Overall the description of the related work and system architecture were clear.

I really appreciated the detailed algorithms (1,2,3) in Section 3.2.   However, there are a few areas where the details were perhaps insufficient.   Section 3 could use a few system diagrams to illustrate how layers are assigned to stages and stages to servers, as well as the client/server caches, what they hold, and when/where their values are sent.
* Section 3.3 load balancing:  How does the system measure throughput?  Continuous updates to DHT?  Is there a notion of total server load (i.e., the amount of memory -- can a new stage even fit on this server?)?
* It isn't clear how consecutive layers are initially assigned to stages.   Is each layer considered in complete isolation?  Doesn't seem so from experiments.   Is it static?  Should you pack the NA onto the fewest, closest servers?
* Do you consider re-locating the client to be centrally located wrt to its servers?   It's essentially a controller node -- it doesn't have to be co-located with the user initiating the task.
* The load-balancing scheme not only allows recovering servers to assign themselves layers, but operational servers to reconsider the layers they serve.   It's possible that the system could easily oscillate as some nodes decide to host new layers, creating a cascade of changes.  Determine if the system is stable and emperically demonstrate.
* You state "they switch layers until the throughput becomes near optimal."  That claim needs to be validated.
* it's great to support pipelining of activations to the next downstream stage and the client.   However it makes failure handling more complicated.  What happens if the client becomes partitioned from the next server but the upstream server does not?  How much does this help performance?

Evaluation
* Really like the microbenchmark in Table 1.  However you state that you simulate failures by resetting pipeline stages.  That might internally be different than Algorithm 1 taking the RPC_Failed path.  In other words, a live server with a "failed stage" has a lot of information and retains connectivity, where a disconnected failed server does not.   So, is re-balancing occuring here or not?   Or are we purely looking at the costs of restart vs. recompute?   I think it's the later, but be clear.
* You run across the wide area and that's great.  But were there any failures?  Were there any re-balancing actions taken? Kill some servers, create some network partitions, create some bad connections -- let's see how Petals really works.
* It's great to compare to a single GPU with offload.  But it seems that running DeepSpeed in the wide area (or even on the emulated network (maybe something like https://www.edge-net.org)) is another important comparison point.
* The model to determine load balancing wasn't evaluated - does it work?

Other items:
* This kind of work is related to different strategies for fault tolerant stream processing.  It might be interesting to see if there are related techniques (checkpointing) and to note what's new here.   Maybe https://raulcastrofernandez.com/papers/sigmod13-seep.pdf is a start.
* Please define block, layer, and stage.  Are they the same, are they different?  Try not to give extra mental work to the readers ;)
* S4.1 listing the three strategies it would be helpful to be clear what "restart" means in each line.   Restart the entire job from the start?  Also, be clear about step.  Is that one step through all stages or a forward step on one stage?
* S3.2 P8 is hard to understand.  Maybe just a bit more?
* S3.2 P7 fix "sends the to the subsequent"
* S3.3 P1 fix "server joins OR leaves" ?
* The language in S3.2 P1 repeats the "every device can act as client and server"



**Strength And Weaknesses:**

Strengths
* Great problem, high impact, fun read
* Technically interesting system architecture with a functional implementation
* Evaluation in the wide area, compelling numbers wrt an offloading baseline

Weaknesses
* The description of the fault-tolerant load-balancing algorithm is insufficient.  The stability of the N-player greedy approach needs to be addressed.  The performance model is not described nor its accuracy evaluated.
* It isn't clear how Petals initially groups layers to the servers or if layer groupings can change.
* The evaluation does not empirically explore actual server failures or network partitions (to the client, between servers).   I.e., there are no results bolstering the core contribution (I comment below on why Table 1 is a great microbenchmark but insufficient in this regard).
* Explores uniform network bandwidth and latency, but it's unclear what this is really testing in the Petals architecture.
* Compares to offloading but should compare to DeepSpeed in the wide area under conditions that illustrate Petals benefits.

**Summary Of The Paper:**

This work presents Petals, a system designed for inferencing and fine-tuning large-language models over distributed, commodity hardware.     This enables users to leverage large, pre-trained models without having exclusive access to high-end hardware, and instead being able to leverage the collective inferencing capability of many systems across the internet.   The system uses pipeline-based model parallelism, distributing model layers across nodes, and both server and client caches to restore state during server failures.  They build the system and evaluate its performance through a failure microbenchmark and illustrate performance on larger clusters with different emulated network conditions.  In that scenario, Petals is about 10x faster compared to offloading on a single GPU.

**Summary Of The Review:**

Interesting, motivated, impactful problem whose solution was evaluated on both local and wide area networks for wide scale inference of LLMs.  The work could use further details and experiments to validate the claim that the system works well under network and node failures.

[Post Author Response] I've reviewed the author's responses and paper changes.  They have addressed a number of my concerns, there remain open questions around system operation at scale (vis-a-vis load balancing).   Some could be addressed by describing the limitations of the current load balancing heuristics and analysis.  Even with these drawbacks, the work could be impactful.  I've adjusted the rating from marginally below to marginally above.

---

> ### Author Response · Authors · 2022-11-14
> **Response to Reviewer 3oHu (Part 2 of 2, Load balancing)**
>
> ### Load balancing
>
> We thank the reviewer for the insightful questions regarding load balancing. We explain its details below (answering your questions along the way).
>
> **Edit (Nov 17):** We updated the paper to include details of the load balancing algorithm (see Appendix D) and **new experiments** validating its properties (Appendix E).
>
> __Measuring throughput__
>
> Before joining for the first time, each server measures its Internet connection throughput (in tokens/sec, using one of public web APIs for doing that) and GPU throughput (in tokens/sec, using a small benchmark running several forward passes). The minimum of these values becomes the overall server throughput, which is then cached for future runs.
>
> __Initial block assignment__
>
> We assume that each server holds a segment of **consecutive** transformer blocks to minimize inference latency. Clients may request forward/backward for the whole segment of blocks or its subsegment, if necessary. Normally, each server loads as many blocks as it can fit in its GPU memory, unless a user limits the number of blocks to utilize the rest of memory for something else.
>
> Before starting, each server calculates the values of $t_i$ - the total throughput of servers currently holding the $i$-th block or loading it (to start holding it in a few minutes). Then, to find the best segment of blocks to serve, the server looks for the most narrow bottleneck in the network. Formally, if the model has $L$ blocks and the server can hold $K$ of them in its GPU memory, we calculate:
>
> $start=\underset{i=1,\ 2,\ …,\ L-K+1}{\arg\min} \mathrm{sorted}([t_i,\ t_{i+1},\ …,\ t_{i+K-1}])$
>
> Here, $\arg\min$ compares the sorted arrays lexicographically and chooses the leftmost $start$ in case of multiple minimums.
>
> This way, the next joining server would always cover a block with the smallest $t_i$. If there are multiple bottlenecks like this, the server will try to cover as many of them as possible (we choose to cover the minimums first because the overall Petals throughput is the minimum of throughputs among model blocks). Among the remaining options, we choose a segment covering as many 2nd minimums as possible, and so on.
>
> __Quality of block assignment__
>
> While the problem of assigning the segments for maximizing throughput does not seem to have a polynomial solution, we have conducted computational experiments and found out that this N-greedy algorithm (running in polynomial time) usually finds an assignment with total throughput of 90-100% of the optimal one (found by trying out all possible assignments in exponential time), given that the values of throughput are realistic to the Petals setup.
>
> __Rebalancing__
>
> Since servers may leave at any time, each server also periodically checks if the current assignment is "good enough" compared to the throughput estimated by running the greedy solution for servers currently present in the network.
>
> Formally, each server periodically looks for a segment of blocks that is more appropriate than the currently loaded blocks with respect to the $\arg\min$ rule above. If it finds one, it simulates how the rest of the servers would behave if we replace the current blocks with the new ones (how the other servers would change their blocks afterwards). If the eventual throughput is at least $p\\%$ better, the server commits to the change and announces that it changes the blocks, then other servers do the rest of the changes (eventually increasing the total throughput).
>
> We use $p=20\\%$ since it gives a reasonable trade-off between the swarm throughput and the frequency of block replacements in our experiments (we don't want to replace blocks too often since this resets attention caches).
>
> __Stability of N-greedy algorithm__
>
> The N-greedy rebalancing algorithm **does not cause oscillations** since a series of block replacements is executed only if it leads to eventually increasing throughput by at least $p\\%$. Once a "good enough" throughput is achieved, servers don't change their blocks anymore (unless an essential number of servers join or leave). We verified this behavior computationally, simulating a network with thousands of servers with different throughputs.
>
> To conclude, the presented heuristic allows servers to quickly close the gaps if a substantial share (maybe 100%) of servers holding certain blocks leave, but avoids excess block replacements otherwise.
>
> ### Other items
>
> > Please define block, layer, and stage.
>
> Both "block" and "layer" refer to a transformer block, i.e. both self-attention and MLP parts with residual and LayerNorm (see Figure 1 in [1]). In turn, "stage" refers to a segment of consecutive blocks located on a specific machine (as in pipeline parallelism). We will clarify these terms in the paper.
>
> We thank the reviewer for suggestions regarding the paper's clarity and will apply them in its future versions.
>
> [1] Vaswani, Ashish, et al. "Attention is all you need." Advances in neural information processing systems 30 (2017).

---

> > ### Comment · Reviewer_3oHu · 2022-11-25
> > **load balancing response**
> >
> > Thanks to the authors for providing detailed feedback and additions to the appendix in the submitted paper.   They have provided clarity and empirical evidence of the load balancing approach.   Ultimately, the load balancing algorithm seems fundamental to the work.  Not only because of varying network and server conditions, but also because it determines initial block assignments.   There remain some questions around the stability of the algorithm, though I believe the heuristics you've employed damp them in the cases in which you've tested.
> >
> > If the paper were to be accepted, the authors should bound the contribution and limits of the current load balancing design, acknowledging gaps in mechanism / experiments.
> >
> > Just some provocations for future work, apologies if I miss any details you've already provided.   In essence, servers coordinate through the single view of the system stored in the DHT.   If many servers attempt to rebalance with the same view, they take the same actions -- nothing prevents them from hosting identical parts.   There is possible evidence of this in Appendix Figure 4 as the number of replacements per minute exceeds the number of total blocks in the model itself (the figure might benefit from showing active server count as well).   Following the algorithm, if k is small, what ensures that all the blocks of the model get stored in a system?  There is also the question of prioritization across models hosted in the same set of servers, which also raises the notion of relative \emph{demand} for each model.  It wasn't clear what timeout is used to damp rebalancing or how expensive a more centralized approach would be.

---

> ### Author Response · Authors · 2022-11-14
> **Response to Reviewer 3oHu (Part 1 of 2)**
>
> We thank the reviewer for the feedback and address their comments below. Concerns related to load balancing are addressed in a dedicated response (see Part 2).
>
> > The evaluation does not empirically explore actual server failures or network partitions (to the client, between servers).
>
> We have, of course, tested PETALS fault tolerance by shutting down servers abruptly. While the system always recovers from server failures, this is a qualitative evaluation (sanity check) that does not fit neatly in Section 4. So, instead, we report a quantitative evaluation of fault tolerance using a smaller controlled setup where we can quickly test for hundreds of failures.
>
> > Explores uniform network bandwidth and latency, but it's unclear what this is really testing in the Petals architecture.
>
> We deliberately run **some** experiments with uniform bandwidth in order to isolate other factors: network failures and latency. However, please note that our real-world experiments have non-uniform latency and bandwidth due to geographically distributed servers (see Section 4.3). The network latency between these systems varied between nodes ranging from 3-5 ms up to 180 ms. We omit the exact server locations for anonymity, but will reintroduce them to the final version of the paper.
>
> ### Questions
>
> > Do you consider re-locating the client to be centrally located wrt to its servers? It's essentially a controller node -- it doesn't have to be co-located with the user initiating the task.
>
> This is indeed a possible optimization for our approach. A centrally located client could indeed help others run their requests faster. However, this could raise security issues, since such a controller would have to execute training / inference code supplied by different nodes. We leave further exploration of this setup to future work.
>
> > What happens if the client becomes partitioned from the next server but the upstream server does not? How much does this help performance?
>
> We agree that this makes fault handling more complicated, but the system remains fault-tolerant. We explore possible solutions in Section 3.2.
>
> In your network partitioning example, the client can still connect to the said server using the circuit relay protocol [1]. Our implementation uses _libp2p_ as a communication layer, which enables automated relaying in such conditions. The performance effects of relays are similar to using a proxy server for Internet connection. That said, clients only resort to this if there are no directly available servers for a given pipeline stage.
>
> [1] https://docs.libp2p.io/concepts/circuit-relay
>
> ### Evaluation
>
> > In other words, a live server with a "failed stage" has a lot of information and retains connectivity, where a disconnected failed server does not. So, is re-balancing occuring here or not? Or are we purely looking at the costs of restart vs. recompute?
>
> In Table 1, we simulate network failures, such as temporary loss of connectivity (by the server) or a failed communication session. As such, no rebalancing occurs in this scenario.
>
> > You run across the wide area and that's great. But were there any failures?
>
> Before running the experiments, we manually tested failures by shutting down intermediate servers. If there are other servers holding these blocks, a client switches to these servers; otherwise, the client will wait until such a stage becomes available.
>
> > It's great to compare to a single GPU with offload. But it seems that running DeepSpeed in the wide area [...] is another important comparison point.
>
> Unfortunately, we observed that DeepSpeed performs extremely poorly in our use case. ZeRO-1 and 2 store full model on every device. As for ZeRO-3, it requires stages to communicate model parameters (~176 GB) on each step, while PETALS only communicates activations (15 KB/token). Thus, the only feasible configuration for DeepSpeed is offloading, which is one of our baselines in Section 4.2. On an unrelated note, we thank the reviewer for referencing EdgeNet.
>
> ### Fault tolerant stream processing
>
> > This kind of work is related to different strategies for fault tolerant stream processing. It might be interesting to see if there are related techniques (checkpointing) and to note what's new here.
>
> This is indeed an interesting parallel. While it is not the main scope of our paper, we agree that there are some interesting parallels between different distributed computing areas. Notably, gradient checkpointing for autograd appears to originate from HPC applications in early 2000s [2], if not even earlier. It is plausible, but not certain, that there are more techniques with common ancestry. However, we believe that this deserves a separate investigation that would not fit neatly in the current work.
>
> [2] Andreas Griewank and Andrea Walther. Algorithm 799: revolve: an implementation of checkpointing for the reverse or adjoint mode of computational differentiation. ACM Transactions on Mathematical Software (TOMS), 26(1):19–45, 2000.

---

### Author Response · Authors · 2022-11-16
**Offloading explained**

Multiple reviewers had questions on how offloading works, so we are adding a small technical reply to alleviate confusion.

Offloading is a procedure that works through the PCIe bus between a NVMe SSD (4 lanes) or a CPU (16 lanes), each lane has a maximum bandwidth of 2 GB/s. So a maximum of 32 GB/s speed.

NVlink is not compatible with offloading as it is a GPU-to-GPU network only.

**Why do we compare Petals with offloading that uses only 1 GPU?**

Counterintuitively, offloading with 2-3 GPUs is **slower** than with 1 GPU. This is because if GPUs are located behind a PCIe switch, the CPU/SSD can only communicate with a single GPU at a time. Thus, the layers are transferred sequentially and not in parallel.

An alternative would be to have GPUs that are not behind the same PCIe switch, which could allow loading parameters in parallel, but this setup has a different bottleneck. Since every GPU loads a part of the model parameters, running the model requires intensive GPU-GPU communication to transfer activations (model parallelism) or parameters (ZeRO). When some GPUs are located behind different PCIe switches, this communication competes for bandwidth with CPU-GPU offloading, slowing down both.

Since running a transformer block on a single token is orders of magnitude faster than copying the block weights from CPU to GPU, one cannot hide the offloading communication cost. As such, the most efficient setup for offloading is a single GPU. Adding a few additional GPUs slows down offloading slightly, unless we reach the point where we can fit the entire model into the GPU memory.

---

> ### Comment · Reviewer_X2vi · 2022-11-22
> **Offloading and NVLink**
>
> I don't understand the claim that NVLink is incompatible with offloading. [1] leverages NVLink GPU-to-GPU communication to parallelize and improve offloading performance. Perhaps, this claim is referring to GPU environments that use PCIe for GPU-to-GPU communication.
>
> [1] ZeRO-Infinity: Breaking the GPU Memory Wall for Extreme Scale Deep Learning

---

> > ### Author Response · Authors · 2022-11-22
> > **NVLink can help with offloading for training, but not inference**
> >
> > We meant that NVlink and offloading are incompatible on a hardware level. The CPU/NVMe can only ingest information from the PCIe lanes, which is not interfaced with NVLink.
> >
> > You are correct that NVLink can help offloading on a software level. Clever partitioning and communication of the offloading state, as done with ZeRO-Infinity, can reduce the requirements of the PCIe bus since communication and computation can be overlapped. However, in the inference case, this is hardly possible because the computation is too insignificant to allow for an overlap of computation and communication.
> >
> > As such, NVLink does not help on a hardware level and helps only on a software level for fine-tuning and training.
> >
> > While according to this argument, a fairer comparison between our fine-tuning setup and offloading would be to use a machine with NVLink and multiple GPUs, it defeats the purpose of our hardware-constrained setup. For most consumers, a 12 GB GPU and 16 GB RAM are the most they can afford. Our work mainly concerns a consumer setting where consumers collaborate together to run large language models. A comparison against an NVLink system would not reflect the environment our algorithm was designed to be used in.
> >
> > Entities that have access to NVLink systems usually also have access to GPU clusters. Our algorithm is designed for communication over the internet and allows the use of large models for users with few resources. The offloading experiment served mainly as a theoretic baseline in the consumer setting where NVLink and multiple GPUs are not available.

---

### Author Response · Authors · 2022-11-16
**Updated paper with clarifications and experiments requested by the reviewers**

We thank the reviewers for taking the time to give feedback about our work.

We uploaded an updated revision of the paper that includes clarifications and experiments requested by the reviewers (new additions are highlighted in green). We summarize the changes below, tagging reviewers who requested them:

- **(3oHu, JnBu)** Added a detailed description of the load balancing algorithm (see Appendix D) and reported experiments validating its properties (see Appendix E).
- **(JnBu)** Reported performance of a server with enough GPU memory to load the entire model (see paragraph 2 of Section 4.1 and Table 2).
- **(3oHu, JnBu)** Added explicit definitions for "model layer", "block", "pipeline stage", and "failure rate" (see footnotes 2 and 6, the caption for Table 1).
- **(3oHu, JnBu)** Added clarifications where requested (paragraph 8 of Section 3.2, the list in Section 4.1) and fixed typos reported by reviewers.

We kindly ask reviewers to take a look at the updated paper and new experimental results and update their feedback accordingly.

---

### Decision · Program_Chairs · 2023-01-20

**Decision:**

Reject

**Justification For Why Not Higher Score:**

Reviewers expressed a few concerns about insufficient evaluation.

**Justification For Why Not Lower Score:**

The system is interesting and the problem is timely.

**Metareview: Summary, Strengths And Weaknesses:**

The paper develops a system for inference and fine-tuning large language models on distributed machines. The system, Petals, includes a few components such as pipeline parallelism, fault tolerance, load-balancing, adapter-based finetuning, quantization and compression, etc.
The system shows improved speed on large clusters, e.g., 10x faster compared to offloading on a single GPU. The reviewers have concerns about the evaluation, such as the evaluation of dynamic load balancing, unfair comparison with single-GPU offloading, comparison with GPUs in a single datacenter, overhead of fault tolerance, and others. The author response addressed some of the concerns.